# OxInflammation Affects Transdifferentiation to Myofibroblasts, Prolonging Wound Healing in Diabetes: A Systematic Review

**DOI:** 10.3390/ijms25168992

**Published:** 2024-08-19

**Authors:** Leonardo L. Silveira, Mariáurea M. Sarandy, Rômulo D. Novaes, Mônica Morais-Santos, Reggiani V. Gonçalves

**Affiliations:** 1Department of General Biology, Federal University of Viçosa, Viçosa 36570-900, Brazil; silveiraleonardo77@gmail.com (L.L.S.); mariaureasarandy@gmail.com (M.M.S.); 2Department of Structural Biology, Institute of Biomedical Sciences, Federal University of Alfenas, Alfenas 37130-001, Brazil; romulo.novaes@unifal-mg.edu.br; 3Department of Animal Biology, Federal University of Viçosa, Viçosa 36570-900, Brazil; 4Animal Science Department, Plants for Human Health Institute, North Carolina State University, North Carolina Research Campus, Kannapolis, NC 28081, USA

**Keywords:** OxInflammation process, wound healing, myofibroblasts, diabetes mellitus, NLRP3, TGF-beta, inflammasome, cytokines

## Abstract

Skin wounds, primarily in association with type I diabetes mellitus, are a public health problem generating significant health impacts. Therefore, identifying the main pathways/mechanisms involved in differentiating fibroblasts into myofibroblasts is fundamental to guide research into effective treatments. Adopting the PRISMA guidelines, this study aimed to verify the main pathways/mechanisms using diabetic murine models and analyze the advances and limitations of this area. The Medline (PubMed), Scopus, and Web of Science platforms were used for the search. The studies included were limited to those that used diabetic murine models with excisional wounds. Bias analysis and methodological quality assessments were undertaken using the SYRCLE bias risk tool. Eighteen studies were selected. The systematic review results confirm that diabetes impairs the transformation of fibroblasts into myofibroblasts by affecting the expression of several growth factors, most notably transforming growth factor beta (TGF-beta) and NLRP3. Diabetes also compromises pathways such as the SMAD, c-Jun N-terminal kinase, protein kinase C, and nuclear factor kappa beta activating caspase pathways, leading to cell death. Furthermore, diabetes renders the wound environment highly pro-oxidant and inflammatory, which is known as OxInflammation. As a consequence of this OxInflammation, delays in the collagenization process occur. The protocol details for this systematic review were registered with PROSPERO: CRD42021267776.

## 1. Introduction

Skin wounds affect a large proportion of the global population and represent an expensive public health issue that affects the entire health system, generating billions of dollars of preventive and treatment expenses, according to the World Health Organization (WHO) [1]. In the wound healing process, there are four steps or events that overlap, which are hemostasis, inflammation, proliferation, and remodeling. Among the most common complications in wound healing are those related to diabetes mellitus (such as vasculopathy and neuropathies), which are especially prevalent and contribute to diabetic ulcer formation [2]. Diabetic ulcers are characterized by a prolonged inflammatory response due to a significant imbalance in cytokine and inflammatory marker expression and the compromised formation and maturation of granulation tissue [3]. Diabetic individuals represent an example of impaired healing, with several impairment factors, including peripheral vascular disease and reduced blood pressure. Uncontrolled and excessive inflammation and, consequently, oxidative stress in the tissue delay the wound healing process [4].

Persistent inflammation has been associated with high levels of IL-1, IL-2, IL-6, IL-17, and tumor necrosis factor (TNF) cytokines, which can result in fibroblast apoptosis and extracellular matrix (ECM) degradation [5,6,7]. During the end of the inflammatory phase, fibroblasts and endothelial cells move into the wound, producing an extracellular matrix and angiogenesis, thereby forming granulation tissue [8]. Many of these fibroblasts acquire the morphological and biochemical appearance of smooth muscle cells, called myofibroblasts [9]. Myofibroblasts, expressing alpha-smooth muscle actin (alpha-SMA), play an essential role in wound healing, mediating growth factor secretion, ECM synthesis, and angiogenesis [10]. In this context, the formation of myofibroblasts occurs when mediators such as platelet-derived growth factor (PDGF), chemokines, and transforming growth factor beta (TGF-beta) bind to their membranes, activating intracellular pathways such as SMAD [11]. Activated receptor complexes mediate canonical TGF-beta signaling by phosphorylating the receptor-regulated effector proteins (R-SMADs) at their carboxy terminals, via the TGF-beta/SMAD signaling pathway. Among the R-SMADs, SMAD2 and SMAD3 mediate the TGF-beta signaling pathway [12] and are responsible for the collagen type I transcription increase in fibroblast cell lines. Therefore, a method to positively target the TGF-beta pathway would be extremely useful in treating diabetic wounds, mainly because it is already known that TGF-beta/SMAD signaling is downregulated in diabetes, leading to the decreased expression and deposition of collagen type I, as well as increased matrix metalloproteinase (MMP) expression [13].

Moreover, TGF-beta can use non-SMAD effectors to mediate some of its biological responses, including non-receptor tyrosine kinase proteins such as Src and FAK, mediators of cell survival (e.g., NF-kB, PI3K/Akt pathways), MAPK (ERK1/2, p38 MAPK, and JNK, among others), and Rho GTPases like Ras, RhoA, Cdc42, and Rac1. Notably, these pathways can also regulate the canonical SMAD pathway and are involved in TGF-beta-mediated biological responses [14]. Some of these pathways are involved in the wound remodeling and closure phase and the release of mediators such as TGF-beta, which stimulate myofibroblast differentiation, which in turn promotes wound contraction and closure [15]. Myofibroblasts expressing alpha-SMA promote contraction and synthesize high protease levels, degrading the extracellular matrix. Thus, beyond their critical role in wound contraction via alpha-SMA expression, myofibroblasts participate in synthesizing extracellular matrix components, influencing wound closure and scars’ mechanical strength [16]. Importantly, during the healing process, the ability of fibroblasts to transform into myofibroblasts is altered in people with diabetes [17]. The versatility of TGF-beta shows that these molecules have a central role in wound healing due to their influence on the inflammatory response and, consequently, the oxidative stress in the tissue. TGF-beta expression can also control angiogenesis, granulation tissue formation, reepithelization, extracellular matrix deposition, and remodeling [18].

The intensified inflammation process is usually associated with increased free radicals and reactive oxygen species (ROS) generation. This process is known as OxInflammation, in which ROS work as secondary messengers, activating inflammation and oxidative stress in positive damage feedback. The lengthy inflammatory phase leads to excessive ROS accumulation, thereby impairing and compromising the entire wound healing process. Excessive ROS generation in diabetes is mainly due to acute increases in serum glucose and oxidative stress generation. Oxidative balance is critical in healing, and the role of ROS can be favorable or deleterious [19]. It is well known that inflammatory and oxidative markers play an essential role in skin wound closure, but little is known about the relationship between inflammatory and oxidative effectors during the transformation of fibroblasts into myofibroblasts in the wound healing process. A comprehensive analysis of the signaling pathways and the relation of the mechanisms involved in oxidative damage and its physiological response has not been systematically undertaken. Furthermore, an overview of the current evidence regarding the advances and limitations of the studies in this field has never been carried out. The development of new therapies, as well as the improvement of gold-standard procedures, is a field that has been heavily explored and described in the literature for the treatment of diabetic wounds. Among the new treatments are negative pressure suction, autologous skin transplantation, and stem cell therapy, as well as advanced drug delivery systems using nanoparticles, hydrogels, liposomes, and others [20,21]. In this context, in this systematic review, we investigated how the excessive presence of reactive oxygen species can affect transdifferentiation to myofibroblasts and compromise the wound healing process in diabetic murine models. In this sense, the data of this study could help to understand the main mechanisms involved in the delay of the wound healing process in diabetic ulcers and provide a guideline for decision-makers or even researchers in developing new products and treatments that can accelerate skin wound closure associated with this comorbidity.

## 2. Methods

### 2.1. Search Strategy

This systematic review was carried out using the Preferred Reporting Items for Systematic Reviews and Meta-Analyses (PRISMA). Its main question was the following: How does excessive ROS’ presence affect transdifferentiation to myofibroblasts and compromise the wound healing process in diabetic murine models? The studies were selected through an advanced search on PubMed–Medline (https://www.ncbi.nlm.nih.gov/pubmed), Scopus (https://www.scopus.com/home.uri), and Web of Science (https://www.webofknowledge.com); both platforms were accessed on 10 April 2021. The protocol details for this systematic review were registered in the Prospective International Registry of Systematic Reviews (PROSPERO: CRD42021267776). For a comprehensive search of relevant papers, the strategy employed was to firstly search the databases for appropriate studies and then search the bibliographic references. PubMed–Medline platform filters were created using the hierarchical distribution of the Medical Subject Headings (MeSH) terms to retrieve the indexed studies. Non-MeSH descriptors were characterized using the Title and Abstract (TIAB) algorithm. A standardized experimental animal filter was applied [22]. The search filters used for the PubMed–Medline search platform were adapted to the Scopus and Web of Science databases, except for the experimental animal filter used in Scopus, which was made available by the website. The complete search strategy is shown in the Appendix A.

### 2.2. Eligibility Criteria

Three independent researchers (LLS, MMS, and MM-S) performed the selection of potentially relevant studies. Initially, the abstracts of all papers recovered in the electronic databases were screened. Duplicate studies were excluded (433 articles), and 826 articles were excluded because the studies were conducted using clinical, ex vivo, and in vitro models. Only in vivo preclinical studies investigating the main mechanisms involved in differentiating fibroblasts into myofibroblasts in excisional wound healing in diabetic models were subjected to the eligibility analysis and considered for inclusion in the systematic review. After initial screening, the full texts of all potentially relevant studies were recovered and evaluated for eligibility. Studies were excluded based on the following criteria: (i) no full text available, (ii) secondary studies (i.e., editorials, commentaries, letters to the editor, and literature reviews without original data). The researchers (RDN, MMS, and RVG) independently analyzed the eligibility criteria, and doubts were resolved by consensus reached through discussion. The inter-rater agreement obtained via our search strategy was evaluated using the kappa coefficient (0.899). Finally, studies were selected that used diabetic murine models with excisional wounds, some of which implemented interventions and some did not, and which reported the performance of myofibroblasts in the healing process. After selecting eligible studies, the indirect screening of the reference lists of all selected studies was performed.

### 2.3. Data Extraction and Management

Considering the detailed characterization of all studies in the systematic review, qualitative and quantitative data were extracted using structured tables. Each table was formulated according to basic methodological requirements used to characterize the studies by descriptive level as follows:

Publication characteristics (author, year of publication, and country of origin);

Experimental model characteristics (animal strain, sex, age, and weight);

Skin wound (type of lesion, site, initial area, number, anesthesia, asepsis, and injury time);

Primary outcome (labeling for myofibroblasts and main cellular pathways activated);

Secondary results (cellularity, synthesis of extracellular matrix components, neo-angiogenesis, cell death, and immunological effectors).

### 2.4. Bias Analysis

The risk of bias was analyzed using the bias risk tool (RoB) from the Center for Systematic Review for Experimentation with Laboratory Animals (SYRCLE, Nijmegen, The Netherlands) and the ARRIVE (Animal Research: Reporting of In Vivo Experiments, London, UK) guidelines. The SYRCLE instrument is based on the Cochrane Collaboration’s RoB Tool, which is adjusted for bias factors that play a specific role in animal intervention studies. The objective was to avoid discrepancies in evaluating the methodological quality in animal experimentation. Signaling questions were answered to facilitate judgment based on the following domains and to increase the transparency and applicability: (i) the sequence generated was randomized; (ii) the blind selection bias of the animals was inadequate; (iii) the blinding of participants and personnel; (iv) the blinding of the outcome assessment; (v) incomplete outcome data; (vi) the data evaluation was complete; (vii) complete results data; (viii) allocation conditions; (ix) intervention data; (x) allocated in groups or individually; (xi) ethics committee data; (xii) data on participant exclusion; (xiii) data on the applied methodology; (xiv) statistical data; (xv) addressed the issue of revision; (xvi) bias due to problems not covered elsewhere in the table. The ARRIVE strategy requires the complete screening of all manuscript sections (abstract to acknowledgments and funding) to evaluate the completeness of scientific reports on animal studies. The screening strategy was based on short descriptions of essential characteristics such as baseline measurements, the sample size, animal allocation, randomization, experimental concealment, the statistical methods, ethical statements, and generalizability. A table summarizing all relevant and applicable aspects was constructed considering the specificity and aims of the systematic review. Individual adherence to bias criteria and the overall mean adherence were expressed as absolute and relative values [23] (Appendix A).

## 3. Results

### 3.1. Publication Characteristics

The initial search obtained 1405 studies, of which 706 were from PubMed–Medline, 260 from Scopus, and 439 from the Web of Science database. Initially, 433 were excluded as duplicates. After reading the titles and abstracts, 829 were excluded, with 143 studies being selected for full reading. Following the reading, 18 studies met the inclusion criteria and were included in the final systematic review. No additional articles were found after reading the bibliographic references of the selected studies, as shown in Figure 1.

The 18 studies selected were from different countries, with China presenting the largest number at six in total (33.33%), followed by the US with three (16.6%); Japan with two (11.1%); and Malaysia, Russia, Korea, Spain, Iran, India, and Germany with one study each, representing 5.55%, respectively (Appendix A). According to the number of articles by year, from 2012 onwards, more studies were described and selected for this research, with a more significant number of studies identified in 2020 (Appendix A). This indicates that, over the last few years, more researchers have investigated closure mechanisms for wounds associated with diabetes, especially regarding the main pathways involved. This increase has led to significant advances and/or improvements in treatments in this field.

### 3.2. Characteristics of Experimental Murine Models

The animals chosen were murine models, including both rats and mice. The most used strains of mice were C57BL/6 in 50% of studies (*n* = 9), while two studies (11.1%) used BKS mice, followed by Swiss albino (5.55%, *n* = 1) and ICR (5.55%, *n* = 1). The rats used included Sprague Dawley in 16.6% of studies (*n* = 3), followed by Wistar (11.1%, *n* = 2). Most studies used males (72.2%, *n* = 13), 22.2% used females (*n* = 4), and one (5.55%) used females and males. All animals used were diabetic; for the majority, this was primarily streptozotocin-induced, namely in 55.5% (*n* = 10), while 38.8% (*n* = 7) were genetically modified. Nine studies (50%) performed further tests to assess glucose levels. Animal ages were reported in 14 studies (77.7%). The ages ranged between 6 and 12 weeks in mice, while, in rats, only one study (5.55%) reported an age of between 8 and 10 weeks. Two studies (11.1%) reported using adult animals, while two others (11.1%) provided no details regarding the animals’ age (Appendix A).

### 3.3. Excisional Diabetic Wound Characteristics

All animals (100%) received dorsal excisional wounds. Asepsis was described in nine of the 18 studies (50%) selected. Sterilization with 70% ethanol was used in almost all studies (*n* = 8, 44.4%) that reported asepsis. All studies reported the excision wound size, ranging from 4 to 20 mm, with 10 mm being the most common size, described in four studies (22.2%), followed by 8 mm, described in three studies (16.6%). Incisional wounds of 4, 5, 6, 7, and 9 mm were performed in two studies (11.1%), and 9, 15, and 20 mm wounds were created in one study each (5.5%). The number of wounds per animal was described in 13 studies (72.2%), ranging from one single wound to six wounds performed in mice and from one to four wounds in rats. Most studies (8–44.4%) reported the use of bandages to prevent catching, biting, and infection, followed by the individual conditioning of the animals (7–38.8%). Three studies (16.6%) reported no care following wounding, and one study (5.5%) only mentioned that no dressing was used.

The time spent observing the wound for further analysis was described in all studies (100%), with seven studies (38.8%) reporting a time period of up to 10 days. In six studies (33.3%), the time was between 11 and 14 days, while five studies (27.7%) adopted a time period of between 15 and 28 days. Twelve studies (66.6%) indicated the anesthetic used for the procedures, with pentobarbital being used in four studies (22.2%) and ketamine and xylazine used in different proportions in mg/kg in three studies (16.6%). The method used for animal euthanasia was described in only two studies (11.1%): one by cervical dislocation and the other by pentobarbital overdose (Appendix A).

### 3.4. Primary Outcomes

#### 3.4.1. Updates and Main Results for TGF-Beta/SMAD Signaling Pathway

In most studies, *n* = 16 (88.8%), skin repair was impaired in diabetic animals, with a concomitant decrease in wound contraction and the reepithelialization rate for healing. TGF-beta played a central role in controlling these pathways, and the most instigated pathway described in this review was the TGF-beta/SMAD (*n* = 14; 77.7%). The results verified in this revision showed that the downregulation of TGF-beta1/SMAD signaling was an essential pathogenic mechanism in wound healing. This mechanism occurs after the phosphorylation of the type I receptor. It specifically recognizes and phosphorylates R-SMADs, including SMAD2 and SMAD3, and the phosphorylation of both destabilizes the interaction with the SMAD anchor for receptor activation (SARA) and increases the affinity for SMAD4 (also called Co-SMAD). This transcriptional complex (R-SMAD/Co-SMAD) is then translocated into the nucleus, where it regulates the transcription of TGF-beta target genes, such as the production of the alpha-SMA protein in the cytoskeleton, responsible for the differentiation of fibroblasts into myofibroblasts [25].

We observed that, in chronic diabetic disease, the TGF-beta I-II receptors are downregulated, with the subsequent absence of the phosphorylation of R-SMADs, which compromises the entire TNF-beta signaling pathway. These alterations were followed by TNF-alpha and MMP-9 upregulation and, consequently, an increase in the inflammatory phase.

#### 3.4.2. Pathways Related to Oxidative Stress and Myofibroblast Differentiation

Most of the studies included in this review also observed increased free radicals and reactive oxygen species (ROS) generation. Some studies named this process OxInflamation, in which the ROS work as secondary messengers, activating inflammation and oxidative stress in positive damage feedback. Other vital pathways related to myofibroblast differentiation were NFK-b, PKC, ROS/Bax-2/Bcl-2/Caspases, and JNK (*n* = 4, 22.2%); each pathway was mentioned separately in one of the studies. Six studies (33.3%) reported the activation of some pathways through ROS’ presence. Increased oxidative stress in diabetic disease resulted in decreased TGF-beta expression and, consequently, an inactive TAK/JNK or PKC/ERK pathway. This reduction inhibited the differentiation and proliferation of genes responsible for fibroblast differentiation into myofibroblasts, delaying the wound healing process.

Five studies (27.7%) reported that excess free radicals and ROS stimulated the expression of pro-inflammatory cytokines such as IL-1 beta, IL-6, and IL-18. The release of these cytokines intensifies the inflammatory process, compromising vascularization at the wound site and decreasing TGF-beta and alpha-SMA factors, which further promotes negative feedback between pro-inflammatory cytokine expression and TGF-beta/SMAD expression.

Some studies also described the nuclear factor kappa beta (NF-κB) pathway as critical in the OxInflammation process, increasing in the wounds in diabetic models. The studies included in this review specifically described the role of NF-κB in regulating the survival, activation, and differentiation of innate immune cells and inflammatory T cells [26]. Moreover, in our review, we observed that the activation of the STAT1, IP3, and NF-κB pathways and membrane receptors like IFN-γR, TNFR, and toll-like in phagocytes, especially macrophages, occurred at the beginning of the wound healing process.

Some studies have also described ROS/Blc-2/Bax-2/Caspase pathway expression. These pathways are involved in the release of calcium into cell cytosol, activating pro-caspases and promoting apoptosis and necrosis in cells. The consequences of these pathways are the decreased expression of genes involved in myofibroblast differentiation and diminished collagen secretion and wound healing in diabetes. In addition, Ca^+2^ mobilization occurs, producing ROS and, consequently, NLRP3 inflammasome activation, resulting in the proteolytic cleavage of Caspase-1, which stimulates IL-1beta and IL-18 maturation, which is responsible for the infiltration of immune cells, vasodilatation, and an endothelial cell response, compromising the performance of TGF-beta and SMAD expression in myofibroblasts.

Therefore, this systematic review revealed how the different pathways are compromised in diabetic models, especially given that the TGF-beta, NF-κB, and NLRP3 molecules are key aspects of the wound healing process, as shown in Figure 2. By deregulating these pathways, the OxInflammation process compromises the entire cascade of events that lead to healing.

### 3.5. Risk of Bias and Methodological Quality Assessments

The bias of reports based on the SYRCLE analysis is detailed in Figure 3 and Figure 4. The original studies adhered to a mean of 50 bias items (Figure 5). According to the methodological analysis criterion, none of the studies met all of these criteria (100%). We can highlight the low risk of bias for incomplete outcome data (attrition bias), selective reporting (reporting bias), and ethical approval. Most studies did not report the sequence generation process. The inadequate concealment of allocation before assignment was not reported in 12 studies (66.6%). No study reported information regarding participants’ knowledge about the interventions (100%) or reported on the understanding of the interventions among the evaluators of the results (100%). The amount, nature, or handling of incomplete results and data reports due to the selective reporting of results and data on wound closure showed no sign of bias (100%). The conditions in which animals were kept were described in 61.1% of studies. Data on any intervention were reported in almost all studies (94.5%). Seven studies (38.8%) did not specify whether the animals were kept in groups or cages. All studies (100%) considered relevant ethical aspects. The methodology used to obtain the results was reported in all cases (100%). Concerning the statistical methods used and the adequacy of these methods, only one study required clarification (5.55%). There was no sign of bias regarding significant populations that were excluded from the studies (100%). Finally, only two studies (11.1%) needed to clarify problems not covered in other parts of the work. All highlighted information is shown in Figure 3 and Figure 4. Therefore, we can conclude that the results of the individual studies show that the current evidence is reliable, due to low levels of bias.

## 4. Discussion

### 4.1. Studies’ Characteristics

The search for therapies that help to heal skin wounds in diabetic models has grown significantly in recent years [45,46,47]. Studies show that skin wound healing is impaired in chronic diseases such as diabetes, showing decreased wound contraction and re-epithelialization [48]. However, there is a lack of knowledge of the main mechanisms and pathways involved in the contraction process of myofibroblasts in diabetic wounds [49]. Our study conducted a careful systematic review to verify and investigate the main mechanisms involved in differentiating fibroblasts into myofibroblasts in the contraction of cutaneous wounds in diabetic murine models. In this regard, we assembled research in this area and critically assessed the quality of the selected studies. Our results provide strong evidence that oxidative stress affects transdifferentiation to myofibroblasts, compromising the wound healing process in diabetes. In addition, it was possible to observe that oxidative stress coexists with inflammation, with a clear overlap in pathways and mechanisms during diabetes. The consequence of this crosstalk is a prolonged wound healing process due to increases in NLRP3 and consequently in IL-1b and IL-18, leading to the intensification of the downregulation of TGF-beta, a critical wound healing pathway. The OxInflammation environment also compromises the SMAD, c-Jun N-terminal kinase, protein kinase C, and nuclear factor kappa beta activating caspase pathways, leading to cell death.

Animal models are an important tool in recreating conditions that allow the investigation of clinical conditions to understand the wound healing process better and test new interventions. The species currently used in in vivo chronic wound models present advantages and limitations. It is known that the natural physiological mechanisms of wound healing in rodents differ from those of humans. In rodents, the *Panniculus carnosus* participates in wound contraction, resulting in its closure, while, in humans, the healing process occurs through re-epithelialization [50]. Therefore, rodent wound models contract and heal at a rate that human skin does not achieve [51]. On the other hand, the porcine model has advantages in terms of its anatomical and physiological similarity to human skin. However, the number of studies that use this animal model is relatively small and variable, making the development of critical studies for systematic reviews difficult. Its more limited use is probably due to it being more costly to purchase, maintain, and keep, as well as requiring specialized handling for anesthetic and surgical interventions [52]. Given this, experimental murine models are widely used to investigate the action mechanisms and pathways related to wound healing, mainly given their low cost, availability, and ease of care and handling, allowing researchers to use a considerable number of animals in experiments, thus producing more reliable results [53]. Additionally, these experimental models are extensively used due to their more similar physiology to that of humans, the possibility of the histological monitoring of the healing process, and the ability to perform macroscopic, biochemical, and biomechanical measurements [54]. Although there are negative aspects to using murine models, the advantages generally outweigh the disadvantages [55].

An important parameter that should be analyzed for wound studies is the wound size and position. Choosing an appropriate location minimizes differences and interference, especially regarding the tensile strength and resistance of the cutaneous tissue [56]. All studies selected in this review performed excisional wounds on the dorsal areas of the animals, being an easily manipulable area that facilitates tissue collection to analyze factors such as wound contraction. Furthermore, it is a place that is difficult for animals to lick when they are properly and individually caged, which is advantageous, as the salivary glands are reservoirs of many rodent growth factors [57].

### 4.2. Transdifferentiation Pathways

According to our review, TGF-beta/SMAD signaling was the most cited pathway, followed by the NLRP3 pathway, being compromised in the diabetic models in the selected studies. TGF-beta is directly linked to the phosphorylation process of the SMAD complex, which, when phosphorylated, is activated and translocated into the nucleus, where it regulates TGF-beta target gene transcription, and ACTA2 gene transcription is activated to increase alpha-SMA protein production in the cytoskeleton. Once the TGF-beta-induced SMAD pathway is activated, numerous feedback mechanisms are activated to modulate the signaling duration and thus stimulate healing [25]. In diabetes, there is the negative regulation of TGF-beta/SMAD and a decrease in type I collagen transcription in fibroblast cell lines and, consequently, a reduction in tissue tensile strength [28]. Moreover, patients with chronic diabetic ulcers were characterized by the downregulation of the TGF-beta I-II receptors and the subsequent absence of the phosphorylation of R-SMAD [58], which compromises the entire TNF-beta signaling pathway. The long-lasting inflammatory phase leads to excessive ROS accumulation, which compromises the overall wound healing process. The TGF-beta pathway regulates the proliferation, survival, apoptosis, and cellular differentiation of many cells. It plays a role in all repair phases and is critical in regulating collagen deposition [23]. In this case, TGF-beta is a candidate mediator of ECM production and remodeling and plays a key role in the deposition and reorganization of the impaired ECM [59]. In this context, a reduction in TGF-beta was observed in diabetic murine models, the most important regulator of alpha-smooth muscle actin expression (alpha-SMA) in myofibroblasts, which impaired the contractile capacity of this cell type [13]. In diabetes, fibroblasts lose the ability to transform into myofibroblasts and differentiation depends on different pathways and mechanisms underlying the role of TGF-beta [17].

Among the pathways involved in this process, we can highlight SMAD, PKC, JNK, and NFK-beta, which are linked to TGF-beta receptor activation. According to Yan et al. [43], an inhibited PKC pathway in diabetics reduced the TGF-beta and alpha-SMA levels and, consequently, fibroblasts’ differentiation into myofibroblasts. Similarly, the JNK pathway is controlled by TGF-beta receptor activation. It controls pro-apoptotic genes that lead fibroblast cells to cell death by apoptosis and necrosis, especially in diabetic models [60]. Additionally, JNK cellular apoptosis pathways can overlap with the ROS/BCL2/Bax 2/Caspase-3 pathways in diabetic models, which can be overexpressed. This promotes cytochrome release, apoptosis pathway activation, and consequently decreased collagen secretion and proliferation capacities, thereby compromising the scar density [44]. On the other hand, the NFK-beta pathway also regulates nitric dioxide (NO2), which macrophages produce during respiratory bursts. Together with IL-1 beta and TNF-alpha, it is involved in inflammatory skin reactions. Importantly, IL-1 beta and TNF-alpha expression are positive in diabetics, which might explain the increased inflammatory process in diabetes [37]. Consequently, there is excessive ROS generation in diabetes, mainly due to acute increases in serum glucose and oxidative stress generation. As such, we can conclude that NF-κB is a transcription factor involved in controlling the expression of several genes linked to the inflammatory and oxidative stress responses and plays a critical role in regulating the survival, activation, and differentiation of innate immune cells and inflammatory T cells [26].

Oxidative balance is critical in healing, and the role of ROS can be favorable or deleterious [19]. As reported by Lin et al. [36], NADPH oxidase 4 (NOX 4) produces free radicals and ROS, especially superoxide and H_2_O_2_, inside phagocytes. The ROS generated from the NOX system mediate pro-inflammatory effects in vascular tissue, and, in phagocytic cells, electrons are transported across the membrane to extracellular oxygen. These superoxides (oxygen) may interact with intracellular messengers, activating redox-sensitive transcription factors such as kappa B and expressing a wide range of adhesion molecules involved in inflammation processes [7]. Different isoforms have been linked to the development of various diseases and disorders, such as cancer, hypertension, stroke, heart failure, neurodegenerative diseases, and diabetes [61]. However, their role is still controversial, and research is ongoing. However, we know that, in diabetes, there is an increase in NOX 4 and ROS. Furthermore, the SMAD pathway, which TGF-beta induces, is inhibited and decreases the activation of myofibroblasts. The excessive ROS generation in diabetes is mainly due to significant increases in serum glucose and oxidative stress. In addition, increased levels of ROS also increase the NLRP3 inflammasome pathways, promoting an OxInflammation environment. Therefore, by decreasing oxidative stress, the inflammatory phase is ameliorated, but this cannot be long-lasting given the decreased levels of IL-1 beta and IL-18 cytokines, increased vascularization due to the release of VEGF, increased TGF-beta presence, and consequently increased alpha-SMA and myofibroblast expression [30].

Considering the OxInflammation environment, different proteins, essential in myofibroblast differentiation, are suppressed in the diabetic model, negatively interfering with wound healing [37]. We can highlight the protein complex NLRP3, which is vital for the inflammatory process. The excessive or altered regulation of NLRP3 inflammasome activity is related to the pathogenesis of a wide variety of inflammatory, autoimmune, and degenerative diseases. There are two signals for NLRP3 inflammasome activation. Signal 1 is triggered by pattern recognition receptor signaling or cytokines, leading to the transcriptional activation of NLRP3 inflammasome components [62], and signal 2 occurs when numerous molecular and cellular events, including excessive mitochondrial ROS production (mtROS), ion flux, and lysosomal damage, are involved in NLRP3 inflammasome activation and enhancement [30]. A better understanding of the molecular mechanisms underlying NLRP3 inflammasome activation will help us to develop prevention and treatment methods for NLRP3 inflammasome-related diseases [62].

Another protein that is compromised in diabetes patients is the proliferating cell nuclear antigen (PCNA) protein, which is involved in DNA replication, repair, and cell cycle regulation. The reduction of this protein negatively affects cell proliferation and decreases angiogenesis and fibroblast growth factor (FGF), known as an essential mitotic factor [27], thereby impairing healing due to the reduced vascularization arising in diabetics [63]. The high metabolic rate in injured tissue requires a good supply of oxygen and nutrients, such that forming new vessels and restoring blood flow is essential for tissue repair [64]. Growth factors were also reported to be inhibited or reduced in diabetics. Fibroblast growth factor (FGF), vascular endothelial growth factor (VEGF), and platelet-derived growth factor (PDGF) act in conjunction with TGF-beta to regulate the migration, proliferation, angiogenesis, and production of the fibroblast ECM [33]. As such, this study focused on mechanisms that influence the differentiation of fibroblasts into myofibroblasts. Here, we observed that the OxInflammation environment plays a crucial role, affecting wound healing in different ways, regardless of the wound size or delays in healing onset. Understanding these pathways is essential to help identify therapeutic targets and the resolution levels involved in regulating the inflammatory response at a clinical level, improving the quality of skin regeneration. It would be appropriate to conduct experimental studies in future research to understand the role of the most important markers of the OxInflammation process, such as 4-hydroxy-2-nonenal (4-HNE) and 8-hydroxy-2′-deoxyguanosine (8-OHdG), and antioxidant enzymes such as catalase, superoxide dismutase, and glutathione, and to understand the whole oxidative cellular chain during the inflammatory and oxidative stress process.

### 4.3. Future Perspectives

Prolonged wound healing can result in increased damage to cells, chronic inflammation, and high levels of ROS, compromising the repair process. Therefore, it is essential to promote effective and rapid healing whenever possible to minimize these adverse effects. Vascular diseases, obesity, burns, and diabetes are the main factors contributing to chronic wounds. The hypoxia response pathway is one of the most important activated pathways, particularly in hyperglycemic patients, resulting in impaired cell migration and vascularization. In this context, understanding the mechanisms activated during wound healing is crucial in designing appropriate treatment strategies to minimize the risk of infection. Despite the availability of several treatment methods, progress in this field is hampered by the limited understanding of the primary mechanisms activated during tissue recovery. These complex mechanisms involve biological processes that must be thoroughly understood to enhance the treatment efficacy. In this context, this review allows an understanding of the ways in which oxidative stress affects transdifferentiation to myofibroblasts and compromises the wound healing process in diabetes. It was found that it is necessary to obtain more information about the mechanisms activated using a preclinical model before the results are translated to the human context. In addition, this systematic review indicates that investing in therapies with high antioxidant potential is necessary to promote the good recovery of skin lesions in diabetes. 

### 4.4. Limitations

Although the systematic review has been listed as a high-level method for the blind evaluation of studies using specific tools, there are some limitations. After reviewing all possible pathways outlined in this review, it is clear that the relationship between inflammation and oxidative stress mechanisms is not well understood. It has been supposed that the process could occur through multiple pathways, but the bias analysis of the in vivo studies also uncovered some underreported information. The primary information not reported included random sequence generation, blinding, and personal and random outcome assessment. The significant heterogeneity, primarily associated with in vivo experimental results, makes it difficult to replicate the work and diminishes the reliability of the research. For example, a lack of information regarding the animal age was identified in most studies, which may reflect a reporting bias as it compromises the report’s quality. Although the individual bias scores were variable, they did not present a temporal influence (year of publication). This finding indicates that the reporting bias was systematically reproduced through the research process, independent of well-known advances in analytical and statistical methods and the increasing availability of guidelines and regulatory strategies adopted to ensure the completeness of the scientific reports in preclinical studies. In all studies included, simple constructs such as experimental blindness, animal allocation and age, sample size calculation, and the rational choice of the administration route were the primary sources of bias. Although these elements are essential sources of intrinsic bias, they also are easily adjustable. The construction of more rigorous experimental designs with acceptable construct validity can be achieved in future studies, especially for translation to the clinical context.

## 5. Conclusions

Based on the data collected from this systematic review, we can conclude that diabetes impairs the transformation of fibroblasts into myofibroblasts, primarily by affecting the expression of growth factors, such as fibroblast growth factor (FGF), vascular endothelial growth factor (VEGF), platelet-derived growth factor (PDGF) and transforming growth factor beta (TGF-beta), compromising fibroblast migration and proliferation, angiogenesis, and the production of the extracellular matrix (ECM), especially collagens I and III. The authors observed that the excessive ROS presence in diabetes promoted an OxInflammation environment, intensifying TGF-beta’s downregulation and NLRP3’s upregulation, which are critical wound healing pathways. The OxInflammation environment also compromised the SMAD, c-Jun N-terminal kinase, protein kinase C, and nuclear factor kappa beta activating caspase pathways, leading to cell death. Therefore, our results provide new insights into how the OxInflammation environment can affect myofibroblasts’ transdifferentiation, delaying wound healing. The findings indicated that the main methodological limitations of the identified studies were based on the recurrent underreporting of the experimental designs, but the development of more comprehensive and controlled studies seems feasible. However, more randomized and controlled studies are required to determine whether and to what extent additional OxInflammation mechanisms can affect transdifferentiation to myofibroblasts, prolonging the wound healing process. In addition, further studies are needed to analyze the role of the critical markers of OxInflammation and better understand the oxidative cellular chain in diabetes, especially in the clinical context.

## Figures and Tables

**Figure 1 ijms-25-08992-f001:**
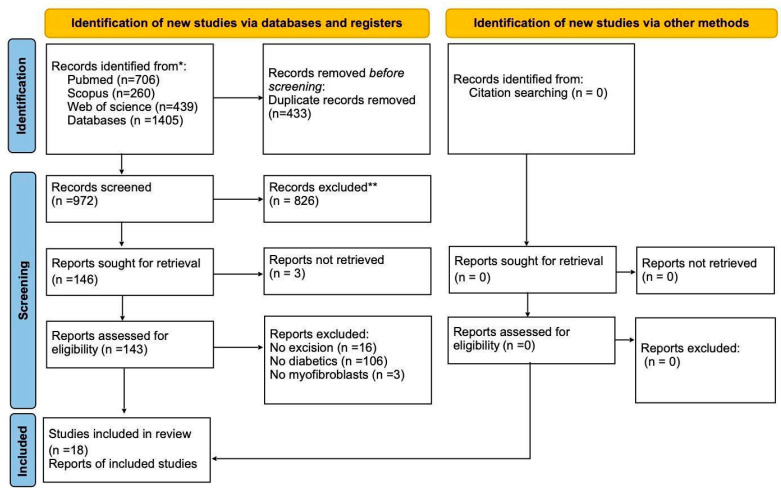
PRISMA diagram. * Consider, when feasible, reporting the number of records identified from each database or register searched (rather than the total number across all databases/registers). ** If automation tools were used, indicate how many records were excluded by a human and how many were excluded by automation tools [24]. For more information, visit http://www.prisma-statement.org, accessed on 17 July 2021. Different phases of the selection of studies to conduct qualitative and quantitative analyses. Flow diagram of the systematic review literature search results. Based on “Preferred Reporting Items for Systematic Reviews and Meta-Analyses: The PRISMA Statement”, http://www.prisma-statement.org.

**Figure 2 ijms-25-08992-f002:**
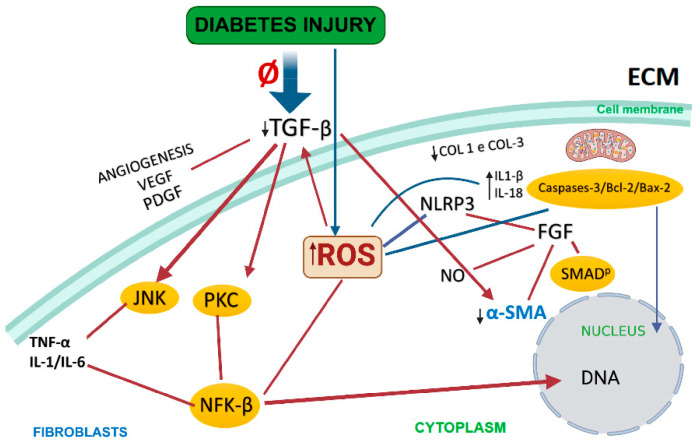
In diabetics, the TGF-β molecule is downregulated, affecting different pathways that promote fibroblast differentiation into myofibroblasts, compromising wound healing. Growth factors VEGF and PDGF, as well as the angiogenesis process, are also severely reduced in diabetics. FGF suppresses α-SMA expression, which, under normal conditions, is activated by the TGF-β molecule. The pathways reported as affected and found in the examined studies are SMAD, JNK, NFK-β, and PKC. During the healing process, when the inflammatory response is accentuated and the ROS levels increase, pathways such as Bcl-2 and Bax-2 are activated, leading to increased Caspase levels and cell death. Therefore, an excessive ROS presence directly affects the pathways studied and compromises healing. Red arrows: inactive myofibroblast pathways. Blue arrow: active. Yellow circles: activated route. ECM = extracellular matrix; TGF-β = transforming growth factor beta; VEGF = vascular endothelial growth factor; PDGF = platelet-derived growth factor; ROS = reactive oxygen species; JNK = c-Jun N-terminal kinase; PKC = protein kinase c; NFK-β = factor nuclear kappa beta; TNF-α = tumor necrosis factor-alpha; ILs = interleukins; COL 1 and COL 3 = collagens 1 and 3, respectively; NLRP3 = NOD, LRR, and pyrin domain-containing protein 3; FGF = fibroblast growth factor; NO = nitric oxide; α-SMA = alpha-smooth muscle actin. Created with BioRender.com.

**Figure 3 ijms-25-08992-f003:**
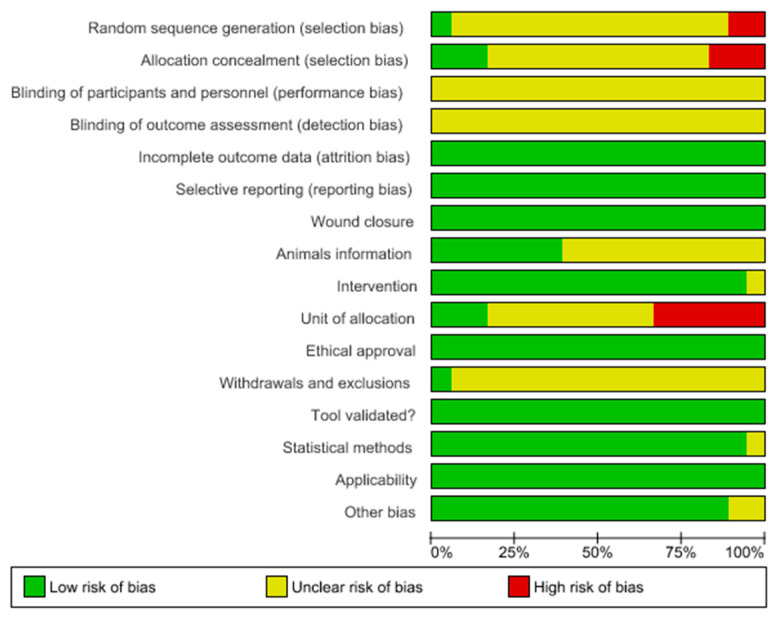
Risk of bias and methodological quality indicators for all studies included in the systematic review using the Systematic Review Center for Laboratory Animal Experimentation (SYRCLE) bias risk assessment. Green: indicating low risk of bias (green); red: indicating high risk of bias; yellow: indicating that the item was not reported, resulting in an unknown risk of bias.

**Figure 4 ijms-25-08992-f004:**
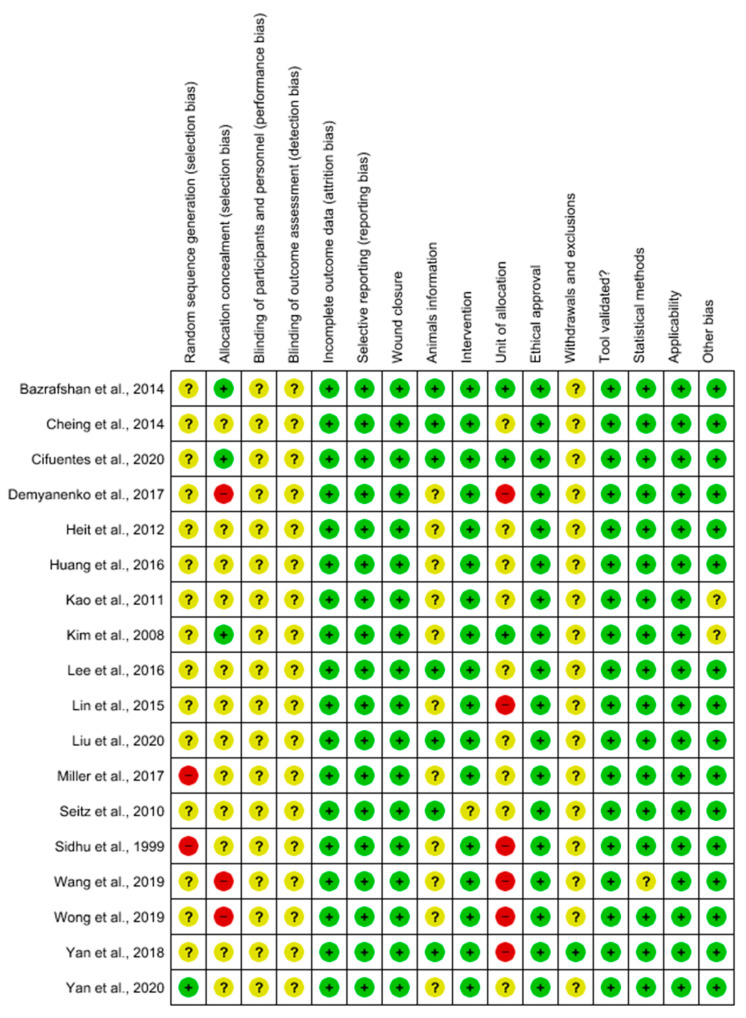
Risk of bias summary: review authors’ judgments about the risk of bias items for each included study. Green: low risk of bias; yellow: unclear risk of bias; and red: high risk of bias. References of the articles in the figure: Bazrafshan et al., 2014 [27]; Cheing et al., 2014 [28]; Cifuentes et al., 2020 [29]; Demyanenko et al., 2017 [30]; Heit et al., 2012 [31]; Huang et al., 2016 [32]; Kao et al., 2011 [33]; Kim et al., 2008 [34]; Lee et al., 2016 [35]; Lin et al., 2015 [36]; Liu et al., 2020 [37]; Miller et al., 2017 [38]; Seitz et al., 2010 [39]; Sidhu et al., 1999 [40]; Wang et al., 2019 [41]; Wong et al., 2019 [42]; Yan et al., 2018 [43]; Yan et al., 2020 [44].

**Figure 5 ijms-25-08992-f005:**
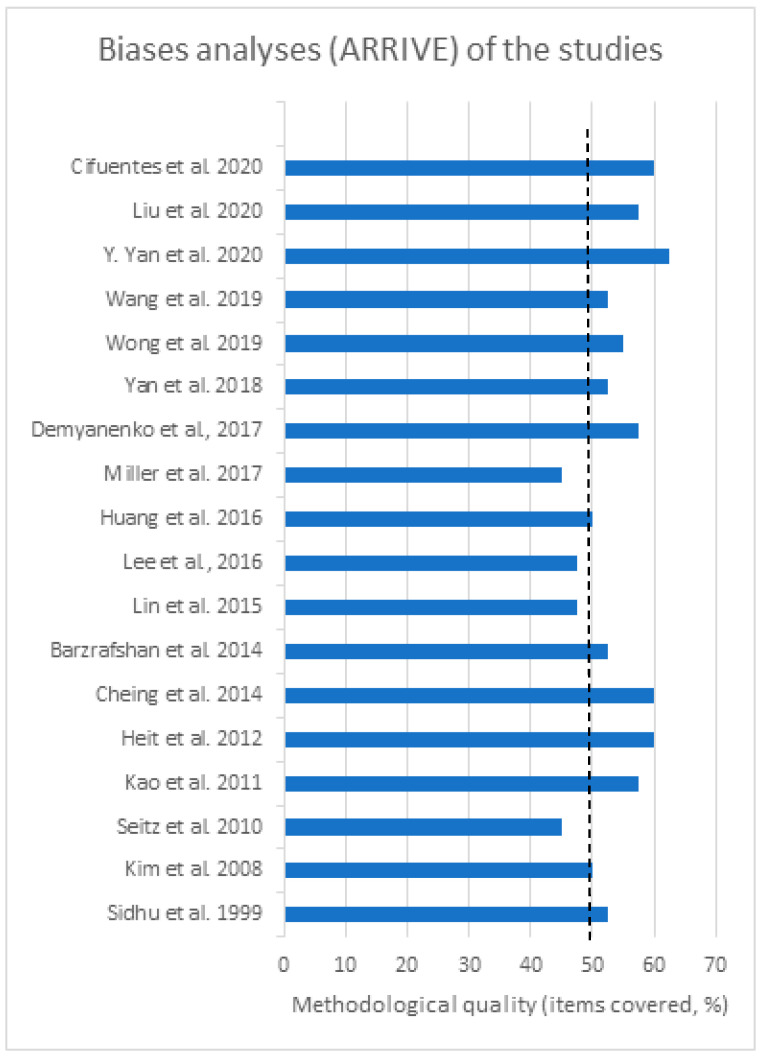
Analysis of methodological bias (reporting quality) for each study included in the review. Based on Animal Research: Reporting of In Vivo Experiments (ARRIVE) guidelines (http://www.nc3rs.org.uk/arrive-guidelines), accessed on 17 July 2021. The dotted line indicates the mean quality score (%). Detailed bias analysis stratified by domains and items evaluated is presented in Appendix A. References of the articles in the figure: Cifuentes et al., 2020 [29]; Liu et al., 2020 [37]; Yan et al., 2020 [44]; Wang et al., 2019 [41]; Wong et al., 2019 [42]; Yan et al., 2018 [43]; Demyanenko et al., 2017 [30]; Miller et al., 2017 [38]; Huang et al., 2016 [32]; Lee et al., 2016 [35]; Lin et al., 2015 [36]; Bazrafshan et al., 2014 [27]; Cheing et al., 2014 [28]; Heit et al., 2012 [31]; Kao et al., 2011 [33]; Seitz et al., 2010 [39]; Kim et al., 2008 [34]; Sidhu et al., 1999 [40].

## Data Availability

The original contributions presented in the study are included in the article/Appendix A; further inquiries can be directed to the corresponding author/s.

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
