# Peer review of "OxInflammation Affects Transdifferentiation to Myofibroblasts, Prolonging Wound Healing in Diabetes: A Systematic Review"

_ijms, 2024, doi:10.3390/ijms25168992_

Round 1

Reviewer 1 Report (Previous Reviewer 3)

Comments and Suggestions for Authors

The authors have intensively revised the manuscript and now it could be recommended for publication in its current form.

Author Response

Not applicable.

Reviewer 2 Report (Previous Reviewer 2)

Comments and Suggestions for Authors

I have no further comments or suggestions. 

Author Response

Not applicable.

Reviewer 3 Report (Previous Reviewer 1)

Comments and Suggestions for Authors

Not applicable

                                   Comments to Authors

The importance of research in delayed wound healing in diabetes has been acknowledged. Although the manuscript is suitable for publication, its quality should be improved. Many grammatical and style errors must be corrected. In addition, the manuscript has the following drawbacks.

1. The title is long and misguided. It should be changed and improved.

2. The abstract is not presented correctly according to the "Instructions for Authors." It must be edited and improved.

3. The introduction is written in such a general form that it must be changed and improved. The last sentence of the introduction part is, "We believe that the results of this study will help to understand the main mechanisms involved in the wound healing process and provide a guideline for decision-makers or even researchers in developing new products and treatments that can accelerate skin wound closure associated with diabetes", however, it is unclear from the introduction which therapeutic approaches for enhancing delayed wound healing in diabetes are known.

Indeed, it is not reasonable that the central question of the research is mentioned in the method section instead of the introduction at first time. The introduction should be clarified in the framework of the analysis of the animal models. In addition, the aims of the research are not presented.

4. "Recovery of research records" is not a suitable title.

5." We observed d that different molecular events are required to transform fibroblasts…" (line 254). It is not clear what the authors mean.

6. Figures 2-5 don't have well-organized and correct titles and captions.

7. " …the current evidence is reliable, and we can reproduce the methodologies and results" (line 347). This statement does not have any scientific value.

8. The discussion section is not presented in due and proper form. For instance, in lines 386-396, this part of the manuscript does not provide any important scientific information relevant to the analyzed results. It is unclear which aspects of the main question of the research are completely clarified and what is necessary to investigate in the future. What can be recommended for future preclinical and clinical studies drawing on the results of this research?

9. The limitation section does not explain the above-mentioned problems either. In addition, the authors don't explain which results are impossible to obtain using the diabetic murine model and which approaches may be practical.

10. The conclusions must be harmonized and improved.

Comments on the Quality of English Language

The importance of research in delayed wound healing in diabetes has been acknowledged. Although the manuscript is suitable for publication, its quality should be improved. Many grammatical and style errors must be corrected. 

Author Response

This manuscript is a resubmission of an earlier submission. The following is a list of the peer review reports and author responses from that submission.

Round 1

Reviewer 1 Report

Comments and Suggestions for Authors

Comments to Authors.

A manuscript by Leonardo L. Silvera et al. attracts great interest. It may be accepted after improving the following drawbacks. The quality of the manuscript should be improved. There are many grammatical errors in the manuscript.

The title is so long. In addition, it is not appropriate to the manuscript's content. It should be changed.

The abstract is not clear and well-organized. It must be edited and improved according to the "Instructions for Authors."

The introduction should be improved because it does not provide sufficient background or represent strong evidence supporting the novelty and importance of the research. The aims of the research should be represented clearly. This part contains no information about products and treatments that can accelerate skin wound healing. In this case, such publications may be recommended:

1. Veith AP, Henderson K, Spencer A, Sligar AD, Baker AB. Therapeutic strategies for enhancing angiogenesis in wound healing. Adv Drug Deliv Rev. 2019 Jun;146:97-125. doi: 10.1016/j.addr.2018.09.010. Epub 2018 Sep 26. PMID: 30267742; PMCID: PMC6435442.

2. Budovsky A., Yarmolinsky L., Ben-Shabat S. (2015). Effect of medicinal plants on wound healing. Wound Repair and Regeneration, 23, 171-183.

3. Patel S, Srivastava S, Singh MR, Singh D. Mechanistic insight into diabetic wounds: Pathogenesis, molecular targets and treatment strategies to pace wound healing. Biomed Pharmacother. 2019 Apr;112:108615. doi: 10.1016/j.biopha.2019.108615. Epub 2019 Feb 20. PMID: 30784919.

The oxInflamation process should be described in the introduction but not in subsection 3.5.

The primary and secondary results are not shown and explained clearly.

The titles and captions of all figures should be edited and improved.

Much information in the discussion section is not appropriate. It should be placed in the results section. The discussion part should be improved because it does not explain the topic according to the "Instructions for Authors." The authors did not pay much attention to wound healing enhancement in various conditions, including diabetes and inflammation. It is unclear which mechanisms are completely researched and which sides require further investigation. What can be recommended for future clinical studies based on this research? There is no analysis of possible future studies and perspectives. In the discussion section, not all conclusions were explained properly. 

Comments on the Quality of English Language

There are many grammatical errors in the manuscript. Minor editing of English required

Author Response

Response to Reviewers’

 ijms-3027909

Dear Reviewer,

Thanks for considering our manuscript for publication in this Journal. Please find below all the corrections made in the manuscript according to the reviewers’
comments. Thanks for the valuable suggestions.

Reviewers’ comments:

Reviewer 1

1- A manuscript by Leonardo L. Silvera et al. attracts great interest. It may be accepted after improving the following drawbacks. The quality of the manuscript should be improved. There are many grammatical errors in the manuscript.

Authors: Thank you for your suggestion. The manuscript was rewritten and revised in native English. Several yellow highlights are throughout the manuscripts, indicating they were all checked.

2- The title is so long. In addition, it is not appropriate to the manuscript's content. It should be changed.

Authors: The title  was changed.

3- The abstract is not clear and well-organized. It must be edited and improved according to the "Instructions for Authors."

Authors: The abstract was rewritten and reorganized

4- The introduction should be improved because it does not provide sufficient background or represent strong evidence supporting the novelty and importance of the research. The aims of the research should be represented clearly. This part contains no information about products and treatments that can accelerate skin wound healing. In this case, such publications may be recommended:

Authors: The introduction was rewritten and the novelty and importance of the research were highlighted. The objective of the study was to understand the main pathways involved in the differentiation of Fibroblasts into myofibroblasts without focusing on the treatment; therefore, I will not add the references in the introduction, but they are exciting, and we will add them in the discussion

  1. Veith AP, Henderson K, Spencer A, Sligar AD, Baker AB. Therapeutic strategies for enhancing angiogenesis in wound healing. Adv Drug Deliv Rev. 2019 Jun;146:97-125. doi: 10.1016/j.addr.2018.09.010. Epub 2018 Sep 26. PMID: 30267742; PMCID: PMC6435442.
  2. Budovsky A., Yarmolinsky L., Ben-Shabat S. (2015). Effect of medicinal plants on wound healing. Wound Repair and Regeneration, 23, 171-183.
  3. Patel S, Srivastava S, Singh MR, Singh D. Mechanistic insight into diabetic wounds: Pathogenesis, molecular targets and treatment strategies to pace wound healing. Biomed Pharmacother. 2019 Apr;112:108615. doi: 10.1016/j.biopha.2019.108615. Epub 2019 Feb 20. PMID: 30784919.

5- The OxInflamation process should be described in the introduction but not in subsection 3.5.

Authors: As suggested, the OxInflammation process was described in the introduction and was deleted to subsection 3.5.

6- The primary and secondary results are not shown and explained clearly.

Authors: The primary and secondary was rewritten and reorganized.

 7- The titles and captions of all figures should be edited and improved.

Authors: The titles and captions of all figures were edited and improved as suggested.

8- Much information in the discussion section is not appropriate. It should be placed in the results section. The discussion part should be improved because it does not explain the topic according to the "Instructions for Authors." The authors did not pay much attention to wound healing enhancement in various conditions, including diabetes and inflammation. It is unclear which mechanisms are completely researched and which sides require further investigation. What can be recommended for future clinical studies based on this research? There is no analysis of possible future studies and perspectives. In the discussion section, not all conclusions were explained properly. 

Authors: We rewrote all the discussion sections, following the instructions for Authors. We paid much more attention to wound healing, bringing the discussion to the clinical studies.

Reviewer 2 Report

Comments and Suggestions for Authors

Major Concerns:

1.      The importance of myofibroblasts in promoting wound contraction is highlighted in the introduction, however, murine models are used in the interrogated studies. In mice and rats, the main mechanism of wound contraction is contraction of the subdermal panniculus carnosus muscle, not via myofibroblasts. It is not possible to know whether any findings are applicable to tight skinned animals, like humans, who lack a panniculus carnosus. Publications using porcine models or murine models with splinted wounds to prevent muscle contraction would have been more favorable to approximate human wound healing, especially regarding the role of myofibroblasts in wound contraction.

2.      As wound healing is an extremely dynamic process, inconsistencies in the original size of the excisional wound (which was not mentioned) and the time point of collection (which ranged between <10 days to 28 days) could be very confounding.

3.      The novel finding of the study appears inconsistent to the background research presented. It is cited that a major problem in diabetic wounds is failure of fibroblasts to transdifferentiate into myofibroblasts (presumably due to dysregulated TGF-β signaling). According to this notion, there are likely few myofibroblasts present in diabetic wounds. However, the suggested novel finding is that OxInflammation is the main pathway activated in myofibroblasts during diabetic wound healing. If there are few myofibroblasts in diabetic wounds, it is unlikely that this finding is biologically meaningful.

4.      At minimum, the above caveats should be discussed in the manuscript and the importance of myofibroblasts in the process of murine wound contraction should be tempered.

5.      The main question of the study (lines 111-113) seems a bit unclear and inconsistent with the main finding. The stated question is: “what are the pathways of activation of myofibroblasts in the wound healing process in diabetic murine models?” Typically, the pathway of myofibroblast activation is via the process of transdifferentiation from a quiescent fibroblast. Is there reason to believe that the TGF-β pathway is not how diabetic myofibroblasts activate? Then, the major finding is that OxInflammation is the main pathway activated in myofibroblasts during diabetic wound healing. As written, this Oxinflammation pathway is downstream of myofibroblast transdifferentiation. Did the current study answer the question it set out to answer? This is very unclear as written.

6.      In the discussion and conclusion, it is not always possible to distinguish what novel finding(s) is gleaned from the current systematic review versus what was already known from the published literature. This should be clarified, and the likely importance of the novel finding should be indicated clearly.

Minor Concerns:

7.      There is an apparent contradiction in the regulation of proteases in the introduction. Lines 73-75 indicate an increase in protease expression in diabetic wounds, while lines 84-85 suggest that myofibroblasts (which it is suggested that diabetic wounds have few of) produce high levels of proteases. Are proteases from diabetic wounds coming from a different source?

8.      The importance of the dorsal location of interrogated wounds is discussed on lines 393-400. It is indicated that the dorsum is a place animals cannot lick. For mice and rats, this is likely untrue. Even if true, is it known that the wounded animals were caged separately post wounding in all studies? Their cage mates may have licked their wounds.

Author Response

Response to Reviewers’

ijms-3027909

Dear Reviewer,

Thanks for considering our manuscript for publication in this Journal. Please find below all the corrections made in the manuscript according to the reviewers’
comments. Thanks for the valuable suggestions.

Reviewers’ comments:

Reviewer 2

The authors described "OxInflammatory responses in the wound healing process: A Systematic review". They conducted a systematic review to investigate the relationship between inflammation and oxidative stress in the skin wound healing process in animal  model. This study should be informative for potential readers especially in wound healing research. I have one suggestion to more improve this manuscript.

1- Shirakami E et al. reported that transforming growth factor beta is a key cytokine that promotes the formation of fibrosis and scarring (Shiraiami E, et al Burns Trauma 2020). However, the authos evaluated only 4 studies ( in Page 5, transforming growth factor β (TGF-β) (n= 4, 14. 4%). Hypertrophic scars are the result of abnormal wound healing. So, they should add more content related to TGF-β in Discussion.

Authors: The TGF-β was rewritten in the discussion. Together, NLRP3 was identified as a key pathway inhibited in diabetes and consequently in the control of the differentiation of Fibroblasts into myofibroblasts. All the modifications are in yellow highlighted in the text.

Reviewer 3 Report

Comments and Suggestions for Authors

In the manuscript, the authors reported a review on the main pathways/mechanisms in diabetic murine models reported in literature and analyzed the advances and limitations in the study on this research topic. Detailed instructions are given for their selection criteria and analysis of literatures. The results presented in the manuscript interesting and comprehensive. The information given could be useful the researchers in the field of study and may be attractive to readers in general.

There are some minor issues found for the authors to consider for revision.

1. The title is not attractive generally

2. Abstract: Lines 24-25, the statement “Our results confirm that diabetes…” seems not appropriate as the authors did not have any experimental results to confirm something stated in the review. It is suggested to revise and make the words mild, such as “On the basis of our analysis, the results generally suggest that diabetes…”

3. Introduction: Line 49, this sentence is not completed and the meaning is not clear.

4. In introduction section, if the authors could give a chart/graph to show how many selected study(ies) is/are published in which year, it could be meaningful to give a generally information/trend on the development of the research field.

5.  Methods: Section 2.2. Eligibility criteria, line 128, “Two independent researchers (LLS, MMS and MMS) performed”, the meaning for this sentence is not clear.

6. Figure 1: the size of the text is too small to read.

7. Discussion: After analyzing the selected literatures, the authors should give comments whether the in vivo experimental results of these studies are generally consistent or any inconsistency found.

8. Conclusion: line 477, “Our results confirm that…”, similar issue as stated in point 2.

Comments on the Quality of English Language

The English needs to be further improved to enhance the clarity. Some sentences are confusing.

Author Response

Response to Reviewers’

 ijms-3027909

Dear Reviewer,

Thanks for considering our manuscript for publication in this Journal. Please find below all the corrections made in the manuscript according to the reviewers’
comments. Thanks for the valuable suggestions.

Reviewers’ comments:

Reviewer 3

In the manuscript, the authors reported a review on the main pathways/mechanisms in diabetic murine models reported in literature and analyzed the advances and limitations in the study on this research topic. Detailed instructions are given for their selection criteria and analysis of literatures. The results presented in the manuscript interesting and comprehensive. The information given could be useful the researchers in the field of study and may be attractive to readers in general.

There are some minor issues found for the authors to consider for revision.

  1. The title is not attractive generally

Authors: The title was changed.

  1. Abstract: Lines 24-25, the statement “Our results confirm that diabetes…” seems not appropriate as the authors did not have any experimental results to confirm something stated in the review. It is suggested to revise and make the words mild, such as “On the basis of our analysis, the results generally suggest that diabetes…”

Authors: These sentences were rewritten, as suggested.

  1. Introduction: Line 49, this sentence is not completed and the meaning is not clear.

Authors: This part of the introduction was corrected.

  1. In introduction section, if the authors could give a chart/graph to show how many selected study(ies) is/are published in which year, it could be meaningful to give a generally information/trend on the development of the research field.

Authors: Thank you for your suggestion. As supplementary material, we added the graph with information about the studies included in this review by each year in the “characteristics of the publications” results.

  1. Methods: Section 2.2. Eligibility criteria, line 128, “Two independent researchers (LLS, MMS and MMS) performed”, the meaning for this sentence is not clear.

Authors: The sentence was corrected.

  1. Figure 1: the size of the text is too small to read.

Authors: The size of the text on the figure was changed.

  1. Discussion: After analyzing the selected literatures, the authors should give comments whether the in vivo experimental results of these studies are generally consistent or any inconsistency found.

Authors: Thank you for your suggestion. We added section 5 describing the limitations and inconsistencies founded in the individual studies, especially related to in vivo experimental results

  1. Conclusion: line 477, “Our results confirm that…”, similar issue as stated in point 2.

Authors: The conclusion was restructured.

Round 2

Reviewer 1 Report

Comments and Suggestions for Authors

None

Reviewer 2 Report

Comments and Suggestions for Authors

No relevant responses were provided from my original review. They appear to be the response to a different reviewer. I expressed this to a member of the editorial team that perhaps the incorrect response for my review was uploaded to my portal, but they indicated there was not an error. I can only assume no thoughtful response to my review was given by the authors. If you believe this to be an error, please rectify with the editorial office.